# Diencephalic Syndrome Due to Optic Pathway Gliomas in Pediatric Patients: An Italian Multicenter Study

**DOI:** 10.3390/diagnostics12030664

**Published:** 2022-03-09

**Authors:** Lucia De Martino, Stefania Picariello, Silvia Triarico, Nicola Improda, Pietro Spennato, Michele Antonio Capozza, Anna Grandone, Claudia Santoro, Daniela Cioffi, Giorgio Attinà, Giuseppe Cinalli, Antonio Ruggiero, Lucia Quaglietta

**Affiliations:** 1Neurooncology Unit, Department of Pediatric Oncology, Santobono-Pausilipon Children’s Hospital, via M. Fiore n° 6, 80129 Naples, Italy; demartinoluci@gmail.com (L.D.M.); stefaniapicariello34@gmail.com (S.P.); 2Department of Woman, Child and General and Specialized Surgery, University of Campania “Luigi Vanvitelli”, via L. De Crecchio n° 2, 80138 Naples, Italy; a.grandone@gmail.com (A.G.); dr.claudiasantoro@gmail.com (C.S.); 3Pediatric Oncology Unit, Fondazione Policlinico Universitario A. Gemelli IRCCS, Largo A. Gemelli, 00168 Rome, Italy; silvia.triarico@guest.policlinicogemelli.it (S.T.); giorgio.attina@policlinicogemelli.it (G.A.); antonio.ruggiero@unicatt.it (A.R.); 4Emergency Unit, Department of Emergency, Santobono-Pausilipon Children’s Hospital, via M. Fiore n° 6, 80129 Naples, Italy; nicolaimproda@gmail.com; 5Section of Pediatrics, Department of Translational and Medical Sciences, University Federico II, via S. Pansini n° 5, 80138 Naples, Italy; 6Pediatric Neurosurgery Unit, Department of Pediatric Neurosciences, Santobono-Pausilipon Children’s Hospital, via M. Fiore n° 6, 80129 Naples, Italy; pierospen@gmail.com (P.S.); giuseppe.cinalli@gmail.com (G.C.); 7Pediatric Oncology Unit, Department of Pediatric Oncology, Santobono-Pausilipon Children’s Hospital, via Posillipo n° 226, 80123 Naples, Italy; micheleantoniocapozza@gmail.com; 8Clinic of Child and Adolescent Neuropsychiatry, Department of Mental and Physical Health, and Preventive Medicine, Università degli Studi della Campania “Luigi Vanvitelli”, via L. De Crecchio n° 2, 80138 Naples, Italy; 9Pediatric Endocrinology Unit, Department of Pediatrics, Santobono-Pausilipon Children’s Hospital, via M. Fiore n° 6, 80129 Naples, Italy; d.cioffi@santobonopausilipon.it

**Keywords:** diencephalic syndrome, optic pathway glioma, NF1, pediatric, chemotherapy, long-term outcomes

## Abstract

Diencephalic syndrome (DS) is a rare pediatric condition associated with optic pathway gliomas (OPGs). Since they are slow-growing tumors, their diagnosis might be delayed, with consequences on long-term outcomes. We present a multicenter case series of nine children with DS associated with OPG, with the aim of providing relevant details about mortality and long-term sequelae. We retrospectively identified nine children (6 M) with DS (median age 14 months, range 3–26 months). Four patients had NF1-related OPGs. Children with NF1 were significantly older than sporadic cases (median (range) age in months: 21.2 (14–26) versus 10 (3–17); *p* = 0.015). Seven tumors were histologically confirmed as low-grade astrocytomas. All patients received upfront chemotherapy and nutritional support. Although no patient died, all of them experienced tumor progression within 5.67 years since diagnosis and were treated with several lines of chemotherapy and/or surgery. Long-term sequelae included visual, pituitary and neurological dysfunction. Despite an excellent overall survival, PFS rates are poor in OPGs with DS. These patients invariably present visual, neurological or endocrine sequelae. Therefore, functional outcomes and quality-of-life measures should be considered in prospective trials involving patients with OPGs, aiming to identify “high-risk” patients and to better individualize treatment.

## 1. Introduction

Diencephalic syndrome (DS) is a rare pediatric disorder characterized by failure to thrive, emaciation, hyperkinesia, despite normal or slightly decreased caloric intake, and normal linear growth. DS commonly occurs in association with low-grade glioma (LGG) arising from the optic pathway, and involving the hypothalamus and the diencephalic structures [1].Optic pathway gliomas (OPG) account for approximately 3–5% of all pediatric brain tumors, and for 20% of brain tumors in very young children. They are more common in patients with neurofibromatosis type 1 (NF1) and represent one of the NF1 diagnostic criteria [2].

OPGs frequently involve the chiasmatic–hypothalamic region, but they can arise anywhere along the optic pathway [3]. Their intimate relation with the optic apparatus, hypothalamus, ventricular system and brain parenchyma results in a constellation of neurological signs and symptoms (nystagmus, hydrocephalus, developmental/neuropsychological disorders, focal neurological deficits), along with endocrinopathies, hypothalamic dysfunction and visual loss.

The majority of cases are pilocytic World Health Organization (WHO) grade I histology, with a smaller proportion being pilomyxoidastrocytomas and gangliomas (WHO grade II) [3]. Although it is a rare occurrence, a more aggressive histopathological diagnosis (diffuse astrocytoma, anaplastic astrocytoma, glioblastoma) can be detected [4,5]. 

LGGs involving the optic pathway are generally slow-growing tumors; thus, their diagnosis might be significantly delayed, with important consequences on treatment outcome and long-term survival. This is particularly relevant to infants and young children presenting with DS as a single sign of disease, who carry the highest risk of morbidity [6]. The deep hypothalamic tumor location is a known risk factor for poor clinical outcome associated with DS onset, whose etiology, although still largely unclear, is strictly related to either an anorexic (inadequate intake/absorption) or a hypermetabolic state (increased energy consumption) [6,7].

The optimal treatment strategy is still a matter of debate [8,9]. Complete surgical resection is indeed typically not feasible due to tumor location, while radiotherapy is usually avoided in young children and those with NF1-related OPGs because of the harmful effect of radiation on the development of the immature brain, radiation toxicity and long-term effects (secondary malignancies, Moyamoya syndrome) [9]. Consequently, chemotherapy is considered the first-line treatment. However, the management of OPGs presenting with DS is challenging, as their prolonged course is characterized by frequent disease progression and multiple lines of therapy [9,10,11,12]. 

Studies evaluating hypothalamic dysfunction and DS associated with OPG are scanty, and several aspects of this condition still need to be fully elucidated. In the present article, we describe a multicenter case series of nine young children with DS associated with OPG, with the aim of providing relevant details about mortality and long-term sequelae associated with this rare condition. 

## 2. Materials and Methods

### 2.1. Study Cohort 

We performed a retrospective study of patients with a radiological and/or histopathological diagnosis of OPG and a clinical diagnosis of DS presenting at 2 Italian tertiary pediatric referral centers (Santobono-Pausilipon Children’s Hospital of Naples, and IRCCS Fondazione Policlinico Universitario A. Gemelli, Università Cattolica Sacro Cuore of Rome) between 2007 and 2021.Diencephalic syndrome was defined as failure to thrive, not explained by vomiting, diarrhea, decreased caloric intake or other causes, and/or a weight <−2 standard deviation score (SDS) with a normal growth rate and/or body mass index (BMI) <−2 SDS [7,13].

Gender, age at diagnosis of OPG, NF1 syndrome diagnosis, histopathology, onset manifestations, anthropometric features, treatment, clinical outcome, number of progressions and duration of follow-up were recorded. OPGs were classified according to the modified Dodge classification (MDC), also known as the PLAN classification [14] (Table 1). 

A neuroradiologist with expertise in this field reviewed magnetic resonance imaging (MRI) scans. Visual assessments were performed by pediatric ophthalmologists and orthoptists in accordance with age-specific test methods. Bilateral vision was categorized according to the World Health Organization (WHO) classification as WHO 0: mild or no visual impairment (≥3/10 or <0.52 Log MAR); WHO 1: moderate visual impairment (<3/10 and ≥1/10 or >0.52 and ≤1 Log MAR); WHO 2: severe visual impairment (<1/10 and ≥1/20 or >1 Log MAR and ≤1.30 Log MAR); WHO 3: profound visual impairment (<1/20 and ≥1/50 or >1.30 and ≤1.69 Log MAR); WHO 4: near blindness (<1/50 and light perception or >1.69 and ≤1.9 Log MAR and light perception); or WHO 5: blindness (no light perception, 0/10 or 2.0 Log MAR).

Weight, height and BMI were expressed as SDS for sex and chronological age, according to Italian reference standards [15]. Obesity was defined as a BMI above 2 SDS according to Italian reference standards [15]. 

Informed consent was obtained from patients, and/or from their parents.

### 2.2. Statistics

Descriptive statistics were used in terms of absolute frequencies and percentages for categorical variables, and the chi-square test or Fisher’s exact test, where appropriate, were applied to compare proportions. Quantitative data were described as the median and range, and differences between groups were assessed with the Mann–Whitney U test. Kaplan–Meier survival curves were calculated for progression-free survival (PFS) and overall survival (OS). In case of no event, patients were censored at the last available follow-up time. In all analyses, *p*-value <  0.05 was considered statistically significant. All analyses were performed using SPSS (version 22.0, SPSS Inc, IBM, Armonk, NY, USA).

## 3. Results

### 3.1. Clinical Features at Diagnosis

We retrospectively identified nine children (six males) with a DS diagnosis, with a median age at onset of 14 months (range 3–26 months). Table 1 summarizes patients’ demographic data, auxological parameters and MDC stage at diagnosis. Four patients had NF1-related OPGs. Patient 8, who has already been described, was affected by NF1 and, in addition, carried a PTPN11 mutation [16]. Children with NF1-related OPG were significantly older than sporadic cases (median (range) age in months: 21.2 (range 14–26) versus 10 (range 3–17); *p* = 0.015). 

All children presented with visual signs, namely, nystagmus, strabismus or optic atrophy. Hydrocephalus requiring cerebrospinal fluid (CSF) diversion was documented in four children (4/9: 43%). Ventriculo-peritoneal shunt was the treatment of choice in three cases, whilst one child underwent endoscopic third ventriculostomy. Three patients (33.3%) had additional neurological signs, such as hypotonia, speech delay and gait disturbances.

Regarding tumor location, the optic chiasm was almost always involved (8/9: 88.9%), followed by the anterior part of the pathway (3/9: 33.3%) and the posterior tracts (3/9: 33.3%) (Table 1). All patients presented hypothalamic involvement. Leptomeningeal metastasis (patient 5) was observed in one case only.

### 3.2. Diagnosis

Six tumors were histologically confirmed as astrocytomas (three pilomyxoid, three pilocytic), while biopsy was not performed in three NF1 patients. Patient 3 underwent surgical debulking because of tumor progression at 4 years of age and was diagnosed with pilocytic astrocytoma. 

BRAF status by RT-PCR was tested in five patients (patient 5, patient 6, patient 7, patient 8, patient 9). Two children with PMX harbored the BRAF-KIAA1549 fusion (patient 9) and BRAF V600E (patient 5). BRAF status was not tested in patient 1 since the analysis was not routinely performed in clinical practice at the time of diagnosis. 

### 3.3. Treatment and Outcomes

Patients were followed for a median time of 8.2 years (range 1–15.2 years). All children received upfront chemotherapy according to the SIOP LGG 2004 protocol, except patient 4 who received the HIT-LGG 1996 protocol (Table 2). Alongside chemotherapy, nutritional support was provided by high-calorie oral nutrition in four patients (44.4%), and by nasogastric tube feeding in five patients (55.6%).

Weight catch-up growth was documented for all patients as early as 6 months after chemotherapy initiation, with BMI normalization within 1 year of treatment (Figure 1). 

Intriguingly, in patient 9, despite tumor progression requiring a second surgical debulking after LGG 2004 standard induction treatment, the body weight continued to increase (+2.1 kg in 9 months). Figure 2 presents MRI scans of patient 9 at varying tumor stages.

MRIs at the end of first-line chemotherapy documented partial response in three cases and stable disease in five. Only one child (patient 9) progressed during first-line treatment, at the end of induction chemotherapy as per the LGG SIOP 2004 protocol (week 24). 

The whole population 5-year OS was 100%, irrespective of tumor histology and treatment (Figure 3). However, all children experienced tumor progression during follow-up within 5.67 years since diagnosis (median time 3.42 years, range 0.5–5.67), with a 5-year PFS of 11.1% (Figure 3), and were treated with several lines of chemotherapy and/or surgery. 

In detail, patient 5 presented with hydrocephalus at 17 months old and was diagnosed with a hypothalamo-chiasmatic PXA BRAF V600E mutation. A vincristine/carboplatin regimen was initiated. The end of treatment MRI demonstrated a reduction in tumor volume. However, six months later, neuroradiological progression was documented (Figure 4). Second-line treatment with dabrafenib and trametinib was subsequently started and is still ongoing.

Although a radiation-sparing approach was adopted at our institutions for such young children, two patients eventually underwent radiotherapy: patient 6 at the age of 9 years because of multiple progressions requiring several tumor debulkings, and patient 7 at the age of 7.2 years since the family rejected fourth-line chemotherapy and additional neurosurgery. 

At the time of last follow-up, all children presented with visual impairment (three blindness WHO 5, one near blindness WHO 4, four mild WHO 0). Neuroendocrine disorders were diagnosed in seven out of nine children (77.8%). Thyroid stimulating hormone deficiency (TSHD) was diagnosed in six children (66.7%), central precocious puberty (CPP) in four (44.4%), growth hormone deficiency (GHD) in two (22.2%) and adrenocorticotropic hormone deficiency (ACTHD) in two (22.2%). One of the two patients that reached pubertal age showed gonadotropin deficiency (GnD). Moreover, three patients (33.3%) had become obese at the last clinical review (Figure 1). Two patients (22.2%) developed permanent central diabetes insipidus (CDI). Five children (62.5%) also displayed neurological and cognitive sequalae, specifically seizures, speech disorders, hemiparesis, learning difficulties and psychomotor delay (Table 2).

During MRI surveillance, patient 8 (NF1 positive and PTPN11 mutated) developed three metachronous LGGs, one of which, in the frontal lobe, was successfully treated with LITT [17]. She was diagnosed with CPP at 3 years of age and treated with a GnRH analogue until the age of 10 years. She is currently 13 years old and presents with mild learning difficulties, hemiparesis and short stature (SDS −3). 

## 4. Discussion

The management of children affected by DS secondary to OPG represents a diagnostic and therapeutic challenge, due to the risk of diagnostic delay and implications for treatment and long-term sequelae. 

The mechanisms underlying DS are still poorly understood and may include increased energy expenditure [18], paradoxically elevated growth hormone levels in response to glucose loads, partial GH resistance and excessive β-lipotropin (a lipolytic peptide) secretion, resulting in increased lipolysis of subcutaneous adipose tissue [6]. However, DS is a frequently neglected cause of failure to thrive in infants and children [19], and, thus, availability of reports may improve medical awareness of this condition as well as knowledge on its pathogenesis. 

In addition, due to the paucity of studies evaluating DS in OPGs [9,18,20,21,22], reports on long-term survivors are important to provide insight into this entity and to guide case management.

Unlike other LGGs, the location of OPGs precludes complete surgical resection due to the risk of incurring unacceptable neurological and functional consequences. Currently, chemotherapy is the mainstay of treatment, resulting in effective and dramatic weight catch-up growth.

To the best of our knowledge, only a few studies have specifically investigated long-term survival in children with DS, and those evaluating the prognostic value of DS at diagnosis in patients with OPG have yielded conflicting results [9,12,22,23]. In our case series, the presence of DS does not seem to impact on mortality rates, with a 5-year overall survival rate of 100%. However, all children in our series experienced at least one neuroradiological or clinical progression and required treatment modification, with a 5-year progression-free survival rate of only 11.25%, in accordance with previous reports [6,9,12]. 

In one of the largest cohorts of DS children with OPG reported thus far [22], Rakotonjanahary et al. found that the risk of death was increased 2.8-fold in children presenting with DS at diagnosis. Moreover, a better prognosis was observed in boys without DS compared with girls with or without DS [22]. Kilday et al. described nine children with DS and OPG treated with chemotherapy. At the time of the last assessment (median follow-up 5.3 years (range 1.2–14.9 years)), all patients were alive, but tumor progression occurred in seven out of the nine DS patients. When compared to all OPG patients aged below 3 years, DS patients progressed earlier than those without DS, although the difference in PFS disappeared over time [9]. 

Regarding the onset age, NF1 children were older than sporadic cases, as previously reported [24]. This might be related to the fact that almost all NF1-related OPGs are pilocytic grade 1 astrocytomas and tend to grow more slowly compared to sporadic cases. However, despite the low growth rate, about half of NF1-related OPG cases become symptomatic in the course of time and need to be treated [24]. In Rakotonjanahary’s series [22], a trend toward a better prognosis was found in NF1 patients during the first years of follow-up, but such a difference completely disappeared over time, resulting in the same OS rate between NF-1 and sporadic cases over a 15-year follow-up.

First-line chemotherapy and supplementation with a high-calorie oral diet or nasogastric feeding rapidly improved the emaciation in our DS cohort within 12 months of treatment, even without evidence of radiological tumor response, and persisted, although tumor progression was seen in patient 9. Our case series also highlights that, despite excellent survival outcomes, young children with OPGs presenting with DS have a poor visual outcome and carry an inexorable risk of additional morbidity, with a considerable prevalence of endocrine and neurological complications. In fact, irrespective of the treatment strategy adopted, hypothalamic involvement itself is a significant predictor of evolving endocrine morbidity and obesity and routinely results in lifelong effects that significantly impact quality of life during and after cure [24]. Although endocrine deficits might be present at the time of diagnosis [24], in our case series, they occurred only after treatment. Hypothalamic dysfunction is a key component of the OPG disease that is often neglected in clinical studies and can be challenging to quantify. The most prevalent endocrine disorder in our patients was TSHD, followed by CPP, CDI, ACTHD and GHD. One patient evolved from CPP to GnD, as previously reported [25]. About half of the patients with long-term follow-up developed hypothalamic obesity after treatment. 

NF1 is characterized by an extremely variable clinical spectrum, ranging from isolated skin manifestations to a more complex multi-system involvement, and approximately 10% of individuals with NF1 share phenotypic features with Noonan syndrome, a RASopathy caused in about half of patients by mutations in PTPN11 [16,26,27]. This was the case with patient 8, who carried an activating PTPN11 variant together with a loss-of-function NF1 variant, resulting in typical NF1 skin findings associated with a severe NF1 neuroradiological phenotype, peculiar cortical hyperintensities and Noonan-like features [16]. Moreover, during follow-up, she developed severe hypostaturalism of mixed etiology. In fact, she showed a blunted pubertal growth spurt, typically observed in both patients with NF1 and Noonan syndrome [28], and the statural outcome was probably worsened by the additive effect of the genetic defects.

This study has two main limitations. The first one is the lack of longitudinal data on visual acuity changes over time. In fact, accurate and reliable visual acuity is very difficult to obtain in children with DS at diagnosis, due to very young age, poor collaboration and severe clinical conditions. However, at the time of the last appointment, all patients had impaired visual function, and none exhibited recovery of neuro-ophthalmic signs (strabismus or nystagmus) during follow-up, indicating a worse visual outcome in patients with DS. 

Secondly, the small number of patients included in the present study does not allow a detailed statistical analysis of predictors of progression and poor clinical outcome. However, also because the condition is rare, the findings of this study offer potentially useful information for this patient population. 

The contribution of biological aberrations merits consideration in future works on this fragile population. In our case series, two patients underwent targeted therapy (dabrafenib + trametinib, and trametinib alone), avoiding radiotherapy or at least delaying it until an older age; however, the relatively short follow-up time does not allow an exhaustive assessment of treatment outcomes and long-term effects of these targeted agents. 

## 5. Conclusions

Optic pathway gliomas are typically treated as a single disease. However, despite an excellent OS, PFS rates are low in young children presenting with DS, leading to repeated chemotherapy cycles and local salvage treatments. As a result, survivors suffer from additional long-term morbidity and continue to struggle with permanent life-changing disabilities, suggesting the need for a better understanding of tumor-induced brain injury. Risk stratification based on prognostic factors such as age at onset, extension along the optic pathway, hypothalamic involvement, histological subtype and tumor biology is important to improve treatment effectiveness and the quality of survival. Therefore, functional outcomes and quality-of-life measures should be considered together with mortality measures in prospective trials involving patients with OPGs, aiming to identify “high-risk” patients and to better individualize treatment [29]. In this context, targeted therapy is an attractive therapeutic approach. MEK inhibitors and BRAF V600E inhibitors have already demonstrated efficacy and are being compared to standard therapy in trials of treatment-naïve patients with LGG [30,31]. However, because these new targeted therapies frequently target genes and pathways that are critical in normal brain development, the acute and subacute long-term sequelae of molecular-targeted therapies need to be carefully monitored in children.

## Figures and Tables

**Figure 1 diagnostics-12-00664-f001:**
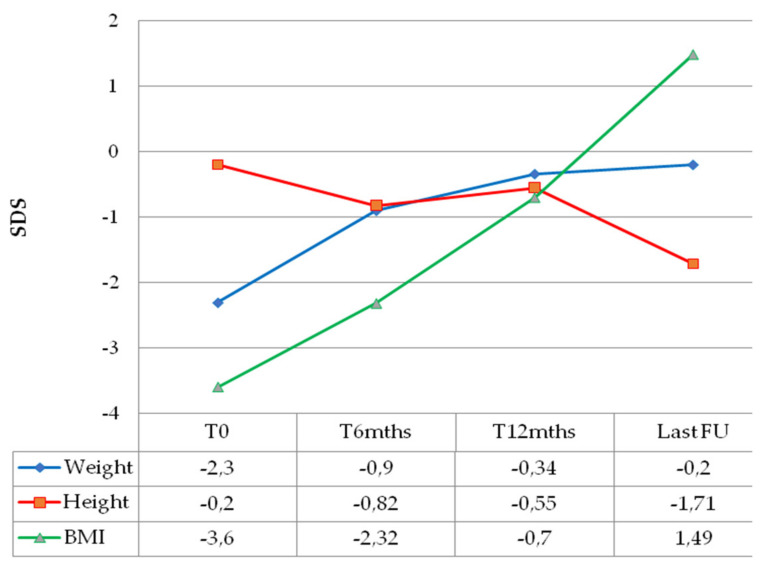
Median anthropometric changes over time of children with optic pathway glioma and diencephalic syndrome.

**Figure 2 diagnostics-12-00664-f002:**
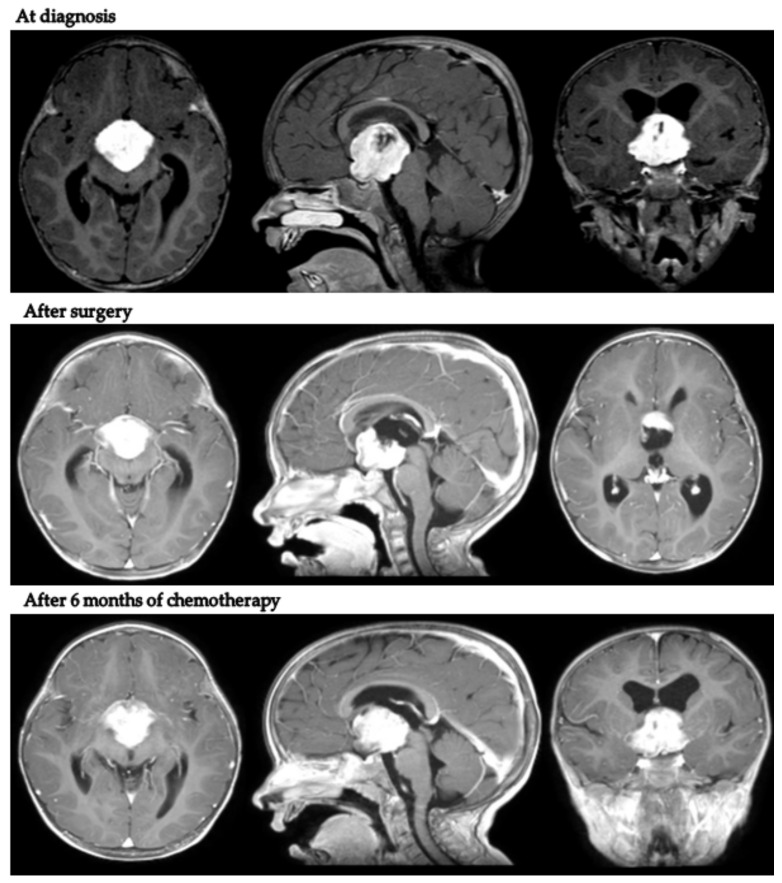
Axial, coronal and sagittal post-contrast T1-weighted MRIs of patient 9 at diagnosis, after neurosurgical biopsy and after the LGG 2004 induction protocol (1st progression).

**Figure 3 diagnostics-12-00664-f003:**
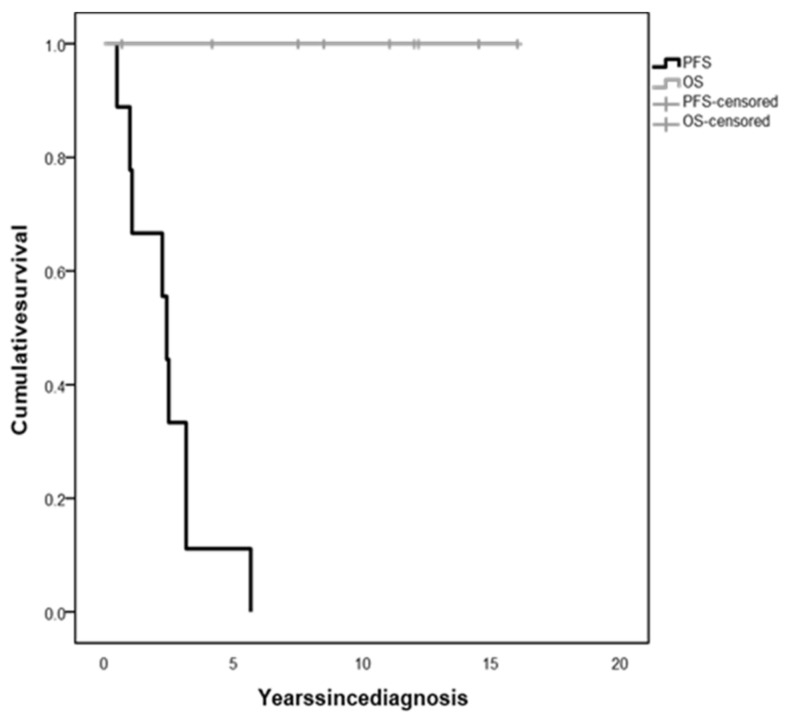
Overall survival (OS) and progression-free survival (PFS) of children with diencephalic syndrome due to optic pathway glioma.

**Figure 4 diagnostics-12-00664-f004:**
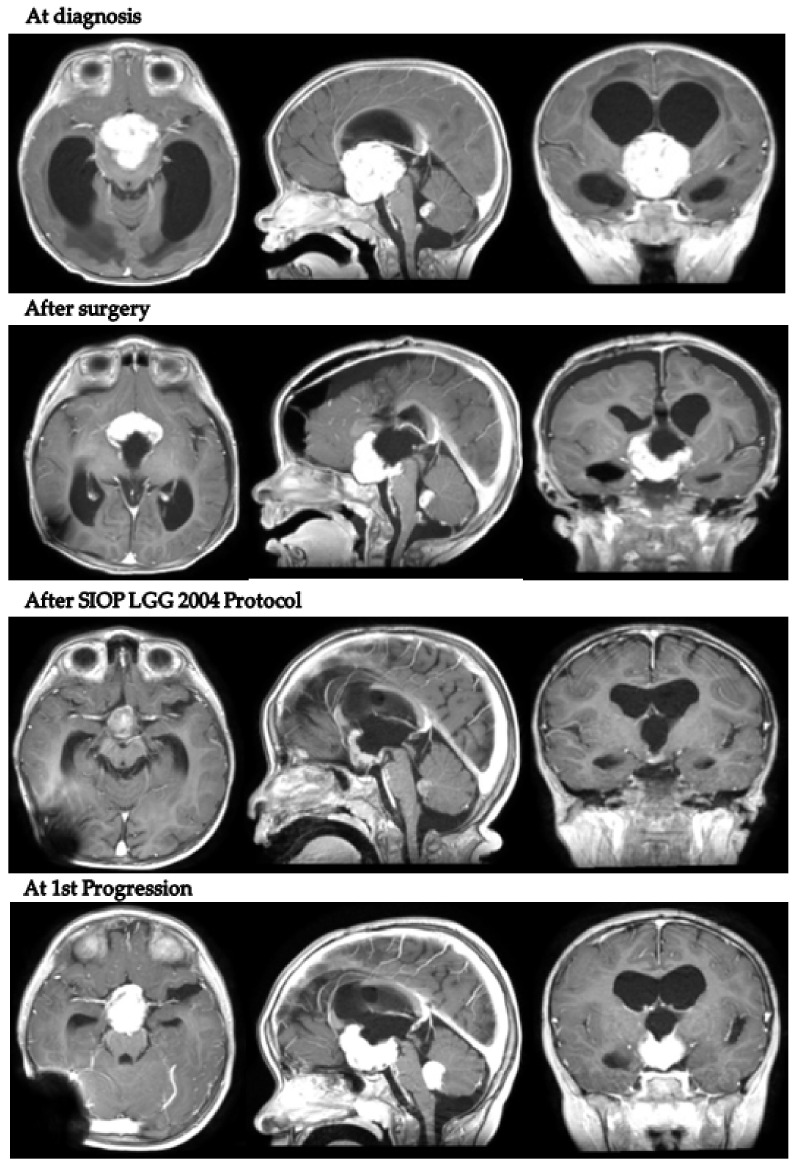
Patient 5 axial, coronal and sagittal post-contrast T1-weighted MRI scans at 4 different timepoints.

**Table 1 diagnostics-12-00664-t001:** Clinical, pathological and biological data for nine cases of diencephalic syndrome related to optic pathway glioma.

	Sex	Age at OPG Diagnosis (Months)	NF1 Status	PLAN Classification	Histology	BRAF Status	Weight (SDS)	Height (SDS)	BMI (SDS)	Hydrocephalus	Additional Signs/Symptoms
Pt1	M	3	Negative	2a 3b H+	PXA	N/A	−3.49	−1.24	−3.94	Yes	Nystagmus
Pt2	F	26	Positive	3b H+	N/A	N/A	−0.76	1.13	−2.05	No	Nystagmus, strabismus, ataxia, speech delay
Pt3	M	21	Positive	2a H+	PA	N/A	−1.32	0.22	−2.17	No	Optic atrophy
Pt4	M	24	Positive	1cL 2bL > R H+	N/A	N/A	−1.25	0.36	−2.01	No	Nystagmus, strabismus
Pt5	M	17	Negative	1b 2a H+ LM+	PXA	V600E mutation	−2.25	−0.80	−2.62	Yes	Strabismus, axial hypotonia, macrocrania
Pt6	M	7	Negative	2bR > L H+	PA	Wild type	−1.57	−0.16	−2.03	Yes	Nystagmus, strabismus
Pt7	M	9	Negative	2a H+	PA	Wild type	−2.82	−0.51	−3.62	No	Strabismus
Pt8	F	14	Positive + Noonan syndrome	1aL 2a 3B H+	PA	Wild type	−4.56	−2.40	−4.76	No	Strabismus
Pt9	F	13	Negative	2aH+	PXA	BRAF- KIAA1549 fusion	−1.95	−0.56	−2.32	Yes	Nystagmus

PXA = pilomyxoid astrocytoma; PA = pilocytic astrocytoma.

**Table 2 diagnostics-12-00664-t002:** Long-term outcomes of patients affected by optic pathway glioma and diencephalic syndrome.

	Patient 1 ^a^	Patient 2 ^a^	Patient 3 ^a^	Patient 4 ^a^	Patient 5 ^b^	Patient 6 ^b^	Patient 7 ^b^	Patient 8 ^b^	Patient 9 ^b^
**Age at last follow-up (years)**	8.42	16.17	7.58	17.17	5.50	12.00	8.75	13.00	1.60
**First-line chemotherapy**	SIOP LGG 2004	SIOP LGG 2004	SIOP LGG 2004	HIT-LGG 1996	SIOP LGG 2004	SIOP LGG 2004	SIOP LGG 2004	SIOP LGG 2004	SIOP LGG 2004
**Number of progressions**	3	2	4	2	1	5	4	1	1
**Chemotherapy at tumor** **progression**	No	N/A	Vinblastine; TMZ	SIOP LGG 2004	Dabrafenib + trametinib	VCR + CPX + cisplatin	VCR + CPX + cisplatin; bevacizumab + irinotecan	VCR + CPX + cisplatin	Trametinib
**Current chemotherapy**	Off therapy 2012	Off therapy 2008	Ongoing	Off therapy 2014	Ongoing	Off therapy 2018	Off therapy 2019	Off therapy 2016	Ongoing
**Radiotherapy**	No	No	No	No	No	Yes	Yes	No	No
**Number of neurosurgeries**	6	1	1	1	3	8	3	3	2
**Year of last neurosurgery**	2014	2006	2016	2005	2019	2018	2019	2019	2021
**Progression time (years)**	3.17	1.08	25	5.67	2.42	3.17	2.5	1.00	0.5
**Follow-up (years)**	8.2	15	5.8	15.2	4	11.3	7.9	11.7	1
**Endocrine** **morbidity**	ACTHD, TSHD, CDI	CPP, TSHD, GHD, GnD, obesity	ACTHD, TSHD, CDI	CPP, TSHD, obesity	None	TSHD, GHD, obesity	CDI, TSHD, CPP	CPP	No
**Neurological outcome**	Psychomotor delay	Normal	Learning difficulties	Normal	Speech disorder	Seizures, psychomotor delay	Normal	Learning difficulties, hemiparesis	Normal
**Visual outcome** **Right eye** **Left eye**	NLP NLP	NLP NLP	NLP LP	NLP NLP	2/10 4/10	1/12 8/10	LP 10/10	4/10 4/10	N/A

^a^ = IRCCS Fondazione Policlinico Universitario A. Gemelli; ^b^ = followed at Santobono-Pausilipon Children’s Hospital of Naples; ACTHD = adrenocorticotropic hormone deficiency; CDI = central diabetes insipidus; CPP = central precocious puberty (CPP); CPX = cyclophosphamide; GHD = growth hormone deficiency; GnD = gonadotropin deficiency; LP = light perception; NLP = no light perception; TSHD = thyroid stimulating hormone deficiency; VCR = vincristine.

## Data Availability

The data that support the findings of this study are available from the corresponding author upon reasonable request.

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
