# Peer review of "Diencephalic Syndrome Due to Optic Pathway Gliomas in Pediatric Patients: An Italian Multicenter Study"

_diagnostics, 2022, doi:10.3390/diagnostics12030664_

Round 1

Reviewer 1 Report

Reviewer's report

Manuscript ID: Diagnostics 1600810
Title: Diencephalic syndrome due to Optic Pathway Gliomas in pediatric patients: an italian multicenter study : an italian multicenter study

Date:2022/2/14

Reviewer's report:
This is an interesting manuscript as it was a study of a highly rare disease - Diencephalic

syndrome due to Optic Pathway Gliomas in pediatric patients: an italian multicenter study

The MS is well prepared, the major limitation is small number of study group (9 pat.) as well as lack of long term follow-up of visual acuity. Nevertheless, its a novel study analyzing the long term follow-up Diencephalic syndrome due to Optic Pathway Gliomas in pediatric patients, which provide us a new perspective toward this rare disease. Therefore, I'm sure this manuscript will add to a growing body of literature in the treatment and evaluation of this rare pediatric tumor. However, there's a few issue need to be answer prior publication.

  1. What was the most characteristic image feature on MRI or CT-scan of this disease ? Please show some important image study of this disease on the manuscript. Pre- chemotherapy and post -chemotherapy image would added more impact to the MS.

  2. How does the tumor response on MRI or CT scan after treatment?

  3. What was the major sign of poor prognosis either clinically or thru  image study ?

Author Response

We thank the reviewer for his comments. Please find below the point by point reply.

  1. What was the most characteristic image feature on MRI or CT-scan of this disease?

Thank you for your question. As stated in the method section, MRI scans were reviewed by an experienced neuroradiologist and tumor location was assessed according to the modified Dodge classification. This classification allows to describe in detail the tumor involvement at multiple anatomical locations and to independently record the hypothalamic involvement as well as the leptomeningeal dissemination (“Radiological classification of optic pathway gliomas: experience of a modified functional classification system”Taylor T, Jaspan T, Milano G, Gregson R, Parker T, Ritzmann T, Benson C, Walker D; PLAN Study Group.Br J Radiol. 2008 Oct;81(970):761-6. doi: 10.1259/bjr/65246351).

Hence, in order to provide a detailed description of the anatomic disease extent along the optic pathway, we  reported the following sentence in the result section: “Regarding tumour location, the optic chiasm was almost always involved (8/9: 88.9%), followed by the anterior part of the pathway (3/9: 33.3%) and the posterior tracts (3/9: 33.3 %) (Table 1). All patients presented hypotalamic involvement. Leptomeningeal metastasis  was observed in one case only (patient 5)."

In addition,for all patients the MDC stage is reported in table 1 in full.

    1. Please show some important image study of this disease on the manuscript. Pre- chemotherapy and post -chemotherapy image would added more impact to the MS.                                                        Thank you for your suggestion. We have added  MRI scans of two cases: Figure 2 and figure 4, showing the neuroradiological course of two suprasellar masses during and after treatment.
  1. How does the tumor response on MRI or CT scan after treatment?

We added the following sentence: “MRIs at the end of first line chemotherapy documented partial response in 3 cases and  stable disease in 5. Only one child (patient 9) progressed during first line treatment, at the end of induction chemotherapy as per LGG SIOP 2004 protocol (week 24).”

  1. What was the major sign of poor prognosis either clinically or thru  image study?

Thank you for raising this point. All patients showed neuroradiological progression.    Unfortunately, the small sample size does not allow us to perform acomprehensive and detailed statistical analysis aimed to identify risk factors of poor prognosis (i.e. Cox Regression analysis of predictors of progression). We have acknowledge this limitationin the discussion section: “Secondly, the small number of patients included in the present study does not allow a detailed statistical analysis of predictors of progression and poor clinical outcome”.

However, we agree with the reviewer that this is a very interesting topic and should be addressed in larger retrospective studies as well as prospective trials.

Reviewer 2 Report

Thia paper presents a case series of 9 patients with diencephalic syndrome. The authors aim to provide information about mortaliry and long-term sequelae by calculating progressive free survival (PFS) and overall survival (OS).  The results showed 1.00 of OS and decreasing PFS according to Fig. 2, although these terms never appear in the Results section.

It seems that the most results of this study are consistent with previous reports as described in the Discusssion section, and the authors share similar viewpoints with previous researchers. Nonetheless, these rere cases presented here may contribute to the future metaanalytic studies.

Author Response

Thank you for your suggestion.

We have added the following sentence in the result section: “Whole population 5-year OS was 100%, irrespective of tumor histology and treatment (Figure 3). However, all children experienced tumor progression during follow-up within 5.67 years since diagnosis (median time 3.42 years, range0.5-5.67) with a 5-year PFS of11.1% (Figure 3) and were treated with several lines of chemotherapy and/or surgery.